Community-based initiatives; health care reform; severe mental illness; health policy; healthcare workers

**Corresponding author:**
Jorge Urrutia Ortiz;
Email: jorge.urrutia@ug.uchile.cl

# Moving psychiatric deinstitutionalization forward: A scoping review of barriers and facilitators

Cristian Montenegro[1,2,3] ⓘ, Matías Irarrázaval Dominguez[2,5],
Josefa González Moller[4], Felicity Thomas[1] and Jorge Urrutia Ortiz[6] ⓘ

[1]Wellcome Centre for Cultures and Environments of Health, University of Exeter, Exeter, UK; [2]Millennium Institute for Research in Depression and Personality, Santiago, Chile; [3]Nursing School, Pontificia Universidad Católica de Chile, Santiago, Chile; [4]Section of Child and Adolescent Psychiatry, Pontificia Universidad Católica de Chile, Santiago, Chile; [5]Department of Psychiatry and Mental Health, Universidad de Chile, Santiago, Chile and [6]Departamento de Psicología, Universidad de Chile, Santiago, Chile

## Abstract

Psychiatric deinstitutionalization (PDI) processes aim to transform long-term psychiatric care by closing or reducing psychiatric hospitals, reallocating beds, and establishing comprehensive community-based services for individuals with severe and persistent mental health difficulties. This scoping review explores the extensive literature on PDI, spanning decades, regions, socio-political contexts, and disciplines, to identify barriers and facilitators of PDI implementation, providing researchers and policymakers with a categorization of these factors. To identify barriers and facilitators, three electronic databases (Medline, CINAHL, and Sociological Abstracts) were searched, yielding 2,250 references. After screening and reviewing, 52 studies were included in the final analysis. Thematic synthesis was utilized to categorize the identified factors, responding to the review question. The analysis revealed that barriers to PDI include inadequate planning, funding, and leadership, limited knowledge, competing interests, insufficient community-based alternatives, and resistance from the workforce, community, and family/caregivers. In contrast, facilitators encompass careful planning, financing and coordination, available research and evidence, strong and sustained advocacy, comprehensive community services, and a well-trained workforce engaged in the process. Exogenous factors, such as conflict and humanitarian disasters, can also play a role in PDI processes. Implementing PDI requires a multifaceted strategy, strong leadership, diverse stakeholder participation, and long-term political and financial support. Understanding local needs and forces is crucial, and studying PDI necessitates methodological flexibility and sensitivity to contextual variation. At the same time, based on the development of the review itself, we identify four limitations in the literature, concerning "time," "location," "focus," and "voice." We call for a renewed research and advocacy agenda around this neglected aspect of contemporary global mental health policy is needed.

## Impact statement

The transition from a mental health system centered on long-term psychiatric hospital care to one centered on community-based services is complex, usually prolonged and requires adequate planning, sustained support and careful intersectoral coordination. The literature documenting and discussing psychiatric Deinstitutionalization (PDI) processes is vast, running across different time periods, regions, socio-political circumstances, and disciplines, and involving diverse models of institutionalization and community-based care. This scoping review maps this literature, identifying barriers and facilitators for PDI processes, developing a categorization that can help researchers and policymakers approach the various sources of complexity involved in this policy process. Based on the review, we propose five key areas of consideration for policymakers involved in PDI efforts: (i) needs assessment, design and scaling up; (ii) financing the transition; (iii) workforce attitudes and development; (iv) PDI implementation and (v) monitoring and quality assurance. We call for a multifaceted transition strategy that includes clear and strong leadership, participation from diverse stakeholders and long-term political and financial commitment. Countries going through the transition and those who are starting the process need a detailed understanding of their specific needs and contextual features at the legal, institutional, and political levels.

## Introduction

Starting during and after World War II in Western Europe and North America, psychiatric deinstitutionalization (PDI) is widely considered a central element of the modernization of

psychiatry. It involves two broad components: (i) the closure or reduction of large psychiatric hospitals and (ii) the development of comprehensive community-based mental health services aiming to promote social inclusion and full citizenship for people living with severe mental illness A broad international consensus supports the need for a shift in mental health care, away from long-term institutionalization and toward comprehensive and integrated community-based and community-shaped services (Campbell and Burgess, 2012; WHO, 2013, 2021a; Thornicroft et al., 2016).

Significant economic, social, and cultural forces have precipitated the development of PDI, including public awareness of the dehumanizing effects of prolonged institutionalization in often poor conditions, the high cost of maintaining large, long-stay institutions, and pharmaceutical developments such as the introduction of psychotropic medication (Turner, 2004; Yohanna, 2013; Taylor Salisbury et al., 2016). For several decades, advocacy movements across the mental health and disability fields have demanded the protection of patients' human rights, including the right to live independently in the community (Hillman, 2005; Mezzina et al., 2019). The UK, Italy, and Finland among other countries are generally regarded as good examples of PDI (Turner, 2004; Westman et al., 2012; Barbui et al., 2018). In the global south, while varying in approach and scale, Brazil, Chile, Sri Lanka and Vietnam have received praise for their efforts to move away from centralized psychiatric institutions (PAHO, 2008; Cohen and Minas, 2017).

Despite the consensus and the declarations by many governments, PDI remains a complex and fragile endeavor. Progress toward PDI varies greatly across and within countries (Hudson, 2019). In some regions, the majority of resources are still invested on centralized, long-term hospitalization (WHO and the Gulbenkian GMHP, 2014); in others, PDI has been delayed with the balance of mental health care shifting in favor of hospital-focused care (Sade et al., 2021); and in other cases, poor management of the PDI process has resulted in tragedy (see e.g., Moseneke's, 2018 account of the Esidimeni tragedy in South Africa).

Understanding the factors that lead to or prevent the transition is crucial to inform the planning and implementation of PDI. Whilst these factors have been documented through the accounts of leaders and experts with hands-on experience, such as in the WHO's Innovation in Deinstitutionalisation report (WHO and the Gulbenkian GMHP, 2014), there has been no previous attempt to systematically scope the literature on barriers and facilitators to PDI.

This paper therefore reports the results of a Scoping Review examining the extent and range of available research regarding barriers and facilitators involved in PDI processes. We organized the specific barriers in seven groups, and the facilitators in six groups, totaling 13 thematic groups. This categorization can be adapted to national realities and different levels of policy action around PDI, to guide research and policy efforts. The synthesis of this information allows us to establish a list of suggestions on ways to move forward.

## Methods

Given that the literature on this topic has not been comprehensively reviewed, the Scoping Review (ScR) (Arksey and O'Malley, 2005) methodology was used. The goal of a ScR is "to map rapidly the key concepts underpinning a research area and the main sources and types of evidence available (…), especially where an area is complex

or has not been reviewed comprehensively before" (Mays et al., 2001, p. 194). For this review, a barrier to PDI was defined as any factor limiting or restricting the transition of care from long-term hospitalization to community-based services and supports. This may include, but is not limited to, issues related to the public-health priority agenda (Shen and Snowden, 2014); challenges in the implementation of mental health services in community settings (Kormann and Petronko, 2004; Saraceno et al., 2007); the resistance of workers employed by psychiatric institutions (Fakhoury and Priebe, 2002); and public and community responses, including stigma, paternalism and other sociocultural factors (Fisher et al., 2005; O'Doherty et al., 2016).

Correspondingly, we define a facilitator as any factor that fosters, promotes, or enables an adequate PDI process. These include the presence of well-organized social activism supporting the rights of persons with mental health problems (Anderson et al., 1998), the acceptance of mental illness as a human condition (Gostin, 2008), service paradigms that enhance social inclusion and citizenship (Fakhoury and Priebe, 2002; Saraceno, 2003) and political willingness (Saraceno et al., 2007).

This ScR was conducted following the Checklist for Preferred Reporting Items for Systematic reviews and Meta-Analyses extension for Scoping Review (PRISMA-ScR) (Tricco et al., 2018). A review protocol was created and registered at the Open Science Platform (doi: 10.17605/OSF.IO/XEBQW). See the protocol and PRISMA-ScR Checklist in Supplementary Materials A and B, respectively.

Three electronic databases were searched in May 2020 – Medline, CINAHL and Sociological Abstracts. Previously published systematic reviews on adults with severe mental health impairment (Lean et al., 2019; Richardson et al., 2019), barriers and facilitators to healthcare access (Adauy et al., 2013) and the deinstitutionalization process (May et al., 2019) informed our search strategy. The strategy combined terms across three dimensions: (i) adults with mental health impairment; (ii) barriers and facilitators related to health care delivery; and (iii) the deinstitutionalization process. The search strategy was not limited by study design or country. Tailored searches were developed for each database (see Supplementary Material C). Eligibility criteria were limited by studies in English and Spanish. All references obtained through the electronic database search and hand search were pooled in EndNote 11 (reference manager) and then uploaded to Covidence (screening and data extraction tool).

Studies selected for inclusion met the criteria detailed in Table 1. Initial eligibility was independently assessed by JU and JG based on title and abstract. At the level of full-text screening, a random sampling of 10% of the selected studies was pilot-tested (with three reviewers) to ensure at least 80% of agreement. Differences in opinions were discussed, and a final decision on their eligibility was made after discussion with CM. A specific data extraction form was created to record full study details and guide decisions about the relevance of individual studies (Table 2). Two reviewers (J.U.O. and J.G.M.) extracted data and checked for accuracy with another reviewer (C.M.C.). Eligibility criteria were further specified to differentiate and exclude specialized substance abuse services involving the legal system. Studies on child institutionalization and substance abuse were also excluded because of the distinct set of causes and challenges associated with these phenomena. Articles related to transinstitutionalization, the transfer of users from psychiatric hospitals to other institutional settings were excluded unless they addressed PDI barriers and facilitators directly.

**Table 1.** Inclusion and exclusion criteria

| | Included | Excluded |
|---|---|---|
| Population | – Studies focused on adult users of long-term mental health services (stays longer than 60 days) | – Studies meeting the above criteria but where participants had a background of a long-term stay in Children Services facilities (children ward, orphans' asylum, group home or residency) or specialized substance abuse services |
| Concept | Studies focused on providers, caregivers (family/friends) and users' account on barriers and facilitators of the psychiatric deinstitutionalization process. Studies focused on PDI processes were included regardless of the study aims. Studies focused on reporting outcome measures related with the community mental health system where only included if they involved a reform process in the context of PDI | – No mention of any facilitator or barrier related to the process of PDI<br>– Studies where the researchers could infer the presence of a barrier o facilitator of PDI but no direct link with PDI processes were clearly set out by the authors were excluded |
| Context | – Studies conducted in mental health setting<br>– No restrictions were placed on the location of intervention delivery (i.e., hospital, day services, community health center, homes) | – No description of the mental health services provided |
| Type of Source | Published and unpublished (gray literature) sources including primary studies, textual papers, technical and governmental reports, calls to action, theoretical and political discussions, historical studies, book chapters and reviews | |
| Language | – Studies wrote in English or Spanish | – All other languages |

*Note:* In the light of the potential differences that may affect the process of deinstitutionalization of Mental Health organizations from Social Services and Specialized Substance Abuse Services (like penal law involvement), this kind of interventions will be excluded.

**Table 2.** Data extraction form

| | |
|---|---|
| Study Information | Correspondence Author |
| | Title |
| | Year of Publication |
| | Country in which the study was conducted |
| | Aim of study |
| | Study Design |
| | Population description |
| | Nº of participants |
| | Setting |
| | Provider type |
| Outcomes | Barriers to Psychiatric Deinstitutionalization |
| | Facilitators to Psychiatric Deinstitutionalization |

During the research process, inclusion criteria adopted a dimensional character, with studies clearly stating barriers and facilitators on one extreme and studies where they had to be inferred, on the other. Given that ScR methodology is defined as an exploratory strategy to map the state of research on a topic (Arksey and O'Malley, 2005; Peters et al., 2015), no attempts were made to assess the methodological quality of the included studies.

Thematic synthesis (Thomas et al., 2004; Lucas et al., 2007; Thomas and Harden, 2008; Harden, 2010) of the selected papers followed a three-stage process. Firstly, it involved free coding the content of the text, to identify barriers and facilitators. Secondly, grouping and organizing the codes into an inductively developed set of categories. Finally, CM examined the categories and their respective codes in the light of the review question to produce an initial set of categories. The match between codes (barriers/facilitators) and categories, and their relevance for the review question was further discussed and refined through rounds of collective revision. A table with examples of the data coding process is available (Supplementary Material D).

To consistently scope the academic production around PDI over several decades, this review includes publications up until May 2020, intentionally excluding the literature related to the Covid-19 pandemic. To properly assess the effects of the Covid-19 pandemic upon processes of Deinstitutionalisation – and on the reality of long-term psychiatric hospitals in general – a different research question, and a tailored design is required.

## Results

The search strategy retrieved 2,250 references. Nine more references were added after hand-searching reference lists and contacting relevant authors. After duplicate removal, 1,915 references were screened by title and abstract, leaving 215 articles for full-text screening. Finally, 52 studies were included in the analysis. Search results and the reasons for excluding full-text articles are provided in the PRISMA flowchart (Figure 1).

### Characteristics of the studies

Included studies were published between 1977 and 2019. This broad temporal scope responds to the fact that an important proportion of research was parallel to the implementation of PDI policies in Europe and the USA during the 1970s and 1980s. Studies were predominantly conducted in the USA (*n* = 22), followed by the UK (*n* = 7) and Canada (*n* = 5). Figure 2 shows an overview of the geographical distribution of the included studies. Regarding the methodology, 25 publications were qualitative studies, 22 were quantitative, and 5 used mixed methods. We provide a summary of the studies' characteristics in Table 3 and descriptions of each study in Table 4.

It is important to consider that this is a general categorization based on the available literature, whose aim is to identify what has been reported as a barrier and as a facilitator in a systematically selected, diverse set of references. We applied thematic analysis to

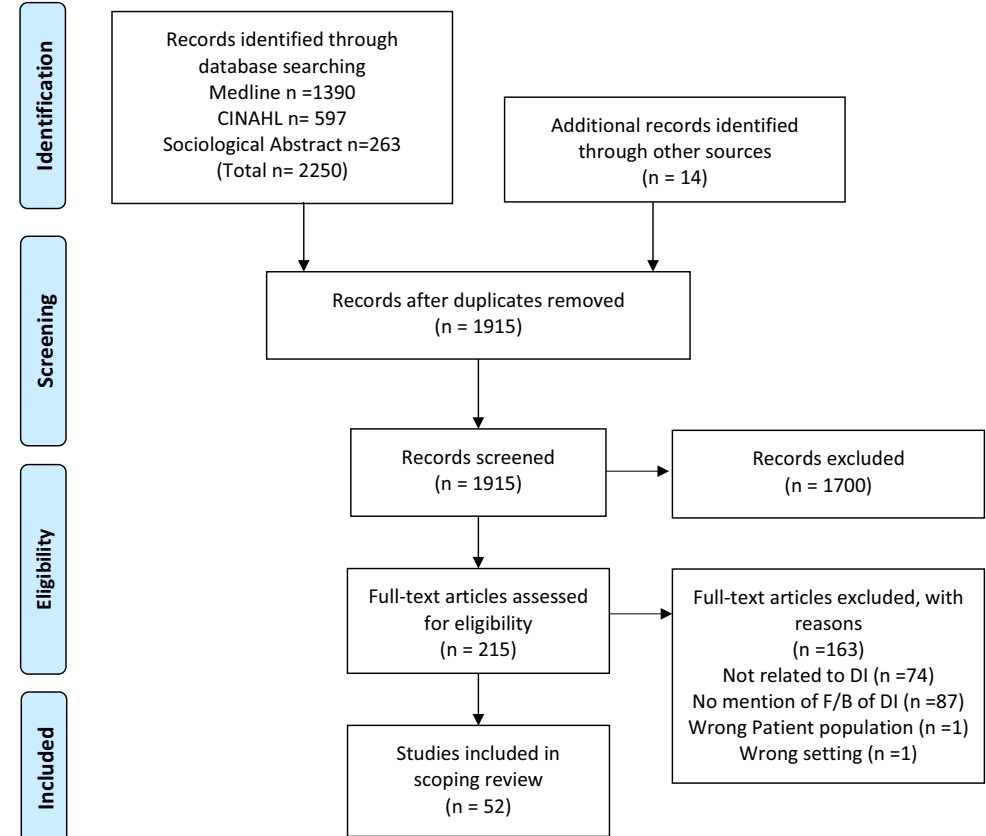

**Figure 1.** PRISMA 2009 flow diagram.

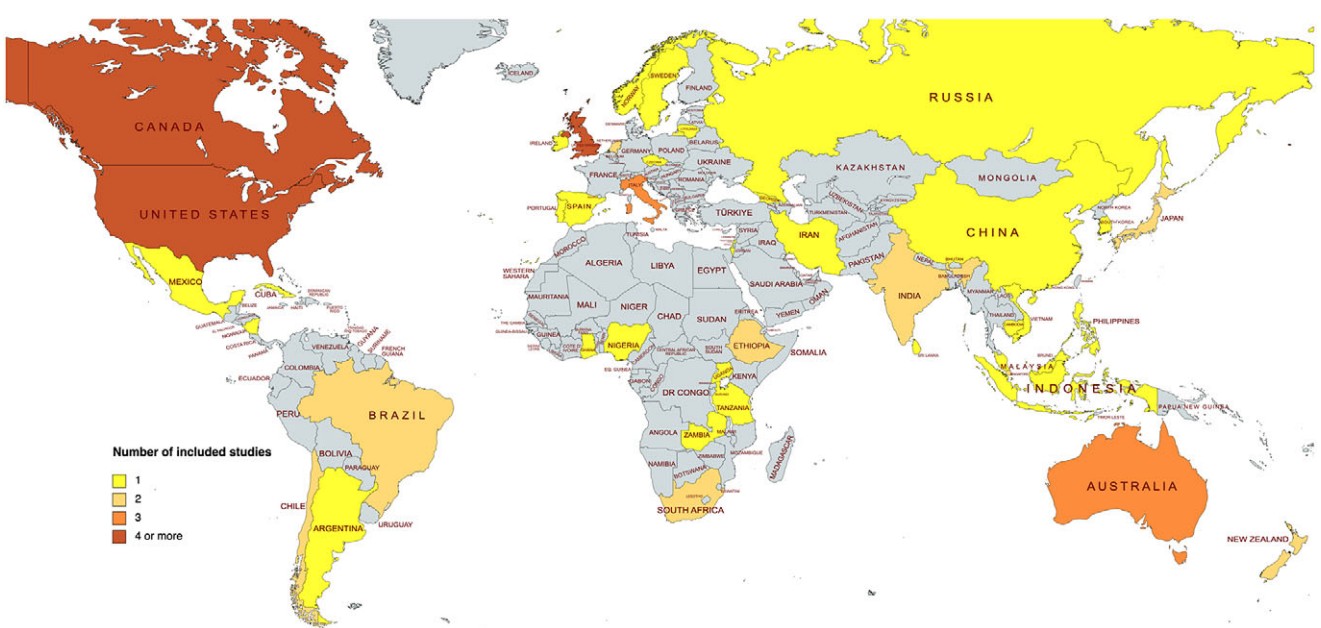

*The following countries were included in one or more multi-country studies: Malaysia, Japan, Ethiopia, Brazil, Nigeria, Uganda, UK, Iran, Italy, Portugal, Cambodia, Philippines, Spain, New Zealand, Usa, Sri Lanka, Chile, India, Republic of Korea, The Netherlandands, Zambia, Indonesia, Tanzania, Singapore, Lithuania, Australia, Georgia, Vietnam, South Africa, Ghana, Sweden, Argentina, Cuba, Jamaica and Mexico.

**Figure 2.** Geographical distribution of included studies.
*Note:* The following countries were included in one or more multi-country studies: Malaysia, Japan, Ethiopia, Brazil, Nigeria, Uganda, UK, Iran, Italy, Portugal, Cambodia, Philippines, Spain, New Zealand, USA, Sri Lanka, Chile, India, Republic of Korea, The Netherlands, Zambia, Indonesia, Tanzania, Singapore, Lithuania, Australia, Georgia, Vietnam, South Africa, Ghana, Sweden, Argentina, Cuba, Jamaica and Mexico.

**Table 3.** Summary characteristics of included studies

|          |                        | Nº of studies |
|----------|------------------------|---------------|
| Setting  | Community mental health | 19 |
|          | Mixed                  | 15 |
|          | Inpatient              | 10 |
|          | Residency              | 4  |
|          | Primary care center    | 2  |
|          | Day Service            | 1  |
|          | Emergency Department   | 1  |
| Provider | Public                 | 30 |
|          | Other                  | 16 |
|          | Private                | 5  |
|          | NGO                    | 1  |
| Language | English                | 52 |
|          | Spanish                | 0  |

the entire set, and on that basis, we developed this initial categorization. We are not establishing the prevalence of each barrier/facilitator across the set or contrasting the characteristics of each barrier/facilitator across regions or within a specific stage in the PDI process. For specific information about the composition of the categories and codes, see Table 5 for barriers and Table 6 for facilitators.

### Barriers to the process of psychiatric deinstitutionalization

Barriers to the process were organized under seven categories, summarized in Table 5 and described in detail below.

#### Planning, leadership and funding

This category includes barriers related to design, implementation, monitoring and overall leadership of the process, and its interaction with other policy processes. One barrier is the lack of accountability from the government to carry out the reform properly, refusing responsibility for housing, social or medical needs and not including other agencies in patient discharge planning (Rose, 1979). The absence of clear operational goals may hinder performance evaluation (Rosenheck, 2000). Charismatic and ideologically driven leadership is important at the beginning, although is vulnerable to political shifts, including elections and changes in government (PAHO, 2008).

Barriers related to funding included the lack of a clear policy that assured the reallocation of resources from hospitals to CMHS (Fakhoury and Priebe, 2002; PAHO, 2008) and a lack of funding to ensure the continuity of community services (Mechanic and Rochefort, 1990; McCubbin, 1994; PAHO, 2008). This is to secure a synchronicity between downsizing psychiatric hospitals and the scaling up of psychosocial interventions.

#### Knowledge/science

Conceptual barriers to promoting PDI were identified. Some authors consider that the lack of research on PDI processes (Bennett and Morris, 1983), paralyze or slow down policy planning and implementation (Shen and Snowden, 2014). At the conceptual level, reducing the concept of community care to narrow geographical proximity can limit the development of community-based interventions (Bennett and Morris, 1983).

Some authors criticized the inadequate transfer and use of certain service paradigms, such as the application of urban-centered interventions to rural locations (Kraudy et al., 1987) without previous identification of rural specificities, creating a disconnection between users and facilities (Schmidt, 2000).

#### Power, interests and influences

Barriers related to the conflict between the interests and perspectives of different groups were grouped under this category.

Authors have discussed the impact of the privatization of mental health care in the wake of the closure of psychiatric hospitals. Market-driven decisions can recreate similar conditions to those in old psychiatric facilities (Rose, 1979). The rise of private hospitals in the United States and their reluctance to participate in non-profit services, such as working with existing public providers, influences access to and the nature of mental health care. Private for-profit hospitals may restrict access to care for uninsured patients (Dorwart et al., 1991). Additionally, private insurance in the United States often encourages unnecessary hospitalization and discourages psychosocial interventions and alternative forms of treatment (Barton, 1983; Freedman and Moran, 1984).

Furthermore, the low cost of hospitalization in some areas, as reported in Asia (Fakhoury and Priebe, 2002), does not provide an economic incentive to push for deinstitutionalization.

The dependence of psychiatric research and development on drug-companies is seen as a barrier. McCubbin stated that the vested interests of the pharmaceutical industry may influence psychiatric practice by selectively supporting medical schools, conferences, and journals, potentially tuning the vision of community mental health into a market opportunity (McCubbin, 1994).

Finally, the lack of relevance of mental health in the political agenda is a crucial, over-encompassing barrier to effective advocacy efforts (Mechanic and Rochefort, 1990; Semke, 1999; PAHO, 2008), as is the uncoordinated and fragmentary nature of these efforts (Mechanic and Rochefort, 1990; McCubbin, 1994; Rosenheck, 2000).

#### Services and support in the community

The slow development of community programmes forced patients to return to long-term institutions, risking chronification (Kaffman et al., 1996). There have been reports of problems caused by the sudden decrease in psychiatric beds without corresponding increases in community-based services. This can result in unintended transfers of patients to other institution-based services and even imprisonment (Shen and Snowden, 2014). Inadequate training of community-based workers, discharge without community support (Shen and Snowden, 2014) and early release promoted by legislatively mandated PDI policies (Kleiner and Drews, 1992) are elements to consider.

The authors identified several barriers to adequate integration of discharged users into their communities, including the absence of jobs and income (Goering et al., 1984), inadequate housing (Grabowski et al., 2009), and insufficient public support (Manuel et al., 2012). Other barriers included challenging behaviors (Allen et al., 2007), old age (Barry et al., 2002), and pessimistic attitudes and feelings of disempowerment and hopelessness among patients (Chopra and Herrman, 2011). In addition, the decrease in disability pensions following an increase in earned income was also identified as a barrier to social integration, as it can discourage work (Chopra and Herrman, 2011).

**Table 4.** Study characteristics of included studies

| References | Country | Aim of study | Setting | Study design | Type of end-user or participant | Number of participants | Provider type |
|---|---|---|---|---|---|---|---|
| Abas et al., 2003 | New Zealand | To describe reasons for admission and alternatives to admission in a government-funded acute inpatient unit | Inpatient Unit | Mixed Methods | Adult patients admitted to a psychiatric hospitalization in the South Auckland Health Mental Health Services | 255 admissions to an acute psychiatric unit in Auckland | Public |
| Aggett and Goldberg, 2005 | UK | To describe the work of a busy Community Mental Health Team with outreach clients. Barriers to collaborative work and some of the team's strategies to overcome them are delineated. | CMHC | Case series | Difficult to engage adult clients between 35 and 52 years old of an outreach community mental health team in an East London borough. | 4 service users | Public |
| Alakeson, 2010 | USA | To examine a range of innovative self-directed care programs in England, Germany, the Netherlands, and the United States. | CMHC | Narrative style | Home and community-based long-term care service users with physical and cognitive disabilities | Inapplicable | Public |
| Allen et al., 2007 | UK | To investigate predictors for out of area placements for people with challenging behaviors and also reports on their costs and basic characteristics. | Mixed | Descriptive Transversal | All people attending to services supporting children and adults with intellectual disability in a large area of South Wales in conjunction with health, education, unitary authority, voluntary and private sector commissioners and providers | 1,458 people | Public |
| Anderson et al., 1998 | USA | To show the changes over 30 years in state institutional populations, interstate variability, movement of individuals into and out of state institutions, costs of state institutional care, and state institution closure as a result of social policy | Inpatient Unit | Descriptive Longitudinal | Patients in institutions for mental disabilities and epileptics between 1950 and 1968 | Inapplicable | Public |
| Ash et al., 2015 | Australia | To describe the implementation of recovery-based practice into a psychiatric intensive care unit, and report change in seclusion rates over the period when these changes were introduced (2011–2013) | Inpatient Unit | Mixed Methods | Consumers (average age 38 years) detained under the SA Mental Health Act. Eleven percent had been charged with or convicted of an offense with a custodial sentence. Common diagnoses were schizophrenia (32%), drug-induced psychosis (18%) and bipolar disorder (manic) (18%). The average length of stay was 11.5 days | 63 people | Public |
| Barry et al., 2002 | USA | To examine the relationship between age, the use of health services and level of functioning in patients with schizophrenia across the adult lifespan | Mixed | Descriptive Transversal | Veterans with schizophrenia drawn from the VA National Psychosis Registry who received a diagnosis of schizophrenia during a VA clinical encounter between 1999 and 2000 | 102.256 | Other: not described |

(*Continued*)

| References | Country | Aim of study | Setting | Study design | Type of end-user or participant | Number of participants | Provider type |
|---|---|---|---|---|---|---|---|
| Barton, 1983 | USA | To discuss the role of mental hospital in the health care system for the elderly | Inpatient Unit | Narrative style | Inapplicable | Inapplicable | Mixed |
| Bennett, 1983 | UK | To describe factors that fostered the deinstitutionalization process in the UK and its consequences in psychiatric services | Mixed | Narrative style | Inapplicable | Inapplicable | Public |
| Bredenberg, 1983 | USA | To present available documentation regarding the implications of residential integration of geriatric ex-mental patients and the well elderly and make recommendations for future action | Residency | Narrative style | elderly discharged mental health service users | Inapplicable | Other: not described |
| Bryant et al., 2004 | UK | To identify how the experience of attending day services met the needs of people with enduring mental health problems | Day Unit | Thematic analysis | patient population | 39 people | Public |
| Chakraborty et al., 2011 | UK | To compare measures of perceived racism, medication adherence and hospital admission between African- Caribbean and white British patients with psychosis | Mixed | Cohort study | participants aged 18–65 years; with a self-assigned ethnicity of Caribbean origin with either parents or grandparents born in the Caribbean; having a Research Diagnostic Criteria-defined psychotic symptom and in receipt of psychiatric services in north London, UK | 110 people | Public |
| Chan and Mak, 2014 | Hong Kong | To examine the mediating role of self-stigma and unmet needs in the relationship between psychiatric symptom severity and subjective quality of life | CMHC | Case series | Adults with schizophrenia spectrum disorders attending community mental health services in Hong Kong | 400 | Nonprofit organization |
| Chopra and Herrman, 2011 | Australia | To assess the long-term outcomes for the original cohort of 18 residents of the Footbridge Community Care Unit (CCU), a residential psychiatric rehabilitation unit at St Vincent's Mental Health Melbourne | ED | Cohort study | 14 schizophrenic and 4 people with schizoaffective disorder | 18 | Public |
| Cohen, 1983 | USA | To clarify conceptions about mental illness in later life and promote the development of mental health services for the elderly in the community | Residency | Narrative style | senior people with mental health difficulties living in housing arrangements | Inapplicable | Other: not described |

(*Continued*)

**Table 4.** (*Continued*)

| References | Country | Aim of study | Setting | Study design | Type of end-user or participant | Number of participants | Provider type |
|---|---|---|---|---|---|---|---|
| Conway et al., 1994 | UK | To report outcomes of community mental health services for people with schizophrenia who had shown very low levels of supported housing and structured day activity | CMHC | Cohort study | patients from West Lambeth, London originally aged 20–65 years who satisfied the research diagnostic criteria for schizophrenia | 51 people | Other: not described |
| Dorwart et al., 1991 | USA | To assess the effect of changes in ownership and types of inpatient settings on the structure of the mental health services system | Inpatient Unit | Analytic transversal | All nonfederal psychiatric hospitals in the United States, including community mental health centers with inpatient units between October 1987 and May 1988 | 915 hospitals | Mixed |
| Evans et al., 2012 | USA | To describe the conversion of partial hospitals into recovery-oriented programs as part of system transformation | CMHC | Narrative style | Stakeholders involved in a transformation of mental health service in a hospital | Inapplicable | Other: not described |
| Fakhoury and Priebe, 2002 | UK | To provide an international overview of deinstitutionalization and review related issues as discussed in the current literature | Mixed | Narrative style | Inapplicable | Inapplicable | Mixed |
| Freedman and Moran, 1984 | USA | To identify and discuss the major policy issues related to the care of the chronically mentally ill, specifically the effects and implications of deinstitutionalization for this particular population | CMHC | Case report | A 32-year-old schizophrenic who has spent more than 10 years in mental health institutions | Inapplicable | Public |
| Goering et al., 1984 | Canada | To describe the 6-month and 2-year postdischarge outcome in each of five aftercare components for 505 subjects in a traditional system of service delivery | Inpatient Unit | Cohort study | Adult people discharged from inpatient units in Toronto | 505 participants | Public |
| Grabowski et al., 2009 | USA | To estimate the cross-state variation in the proportion of nursing home admissions indicating a mental illness, and the proportion of persons with mental illness admitted to nursing homes | Residency | Descriptive Transversal | Nursing home admissions in the USA during 2005 | 1.150.734 new admissions | Private |
| Huang et al., 2017 | Singapore | To design a general practitioner–partnership programme in an institute in Singapore to facilitate the transition to community services and gauge the impact of the interventions chosen to improve uptake of referrals | CMHC | Mixed Methods | Stable mental health service users referred to the GP from December 2014 to January 2016partnership programme in a mental health institute in Singapore | 238 service users | Private |
| John et al., 2010 | India | To describe the successful management of a person with schizophrenia in the community through a primary care team in liaison with psychiatrist services | CMHC | Case report | adult with psychotic symptoms living in an urban area of India | 1 person | Public |

**Table 4.** (*Continued*)

| References | Country | Aim of study | Setting | Study design | Type of end-user or participant | Number of participants | Provider type |
|---|---|---|---|---|---|---|---|
| Kaffman et al., 1996 | Israel | To report on an alternative community care program that has been developed and implemented in the Kibbutz Clinic for the treatment and rehabilitation of the severely mentally ill | CMHC | Mixed Methods | adult people with a severe mental illness with poor functioning who participated in the program conducted in Telem, Israel, for at least 18 months and followed up for a minimum of 4 years | 124 patients | Private |
| Kalisova et al., 2018 | Czech Republic | To assess the effect of the S.U.P.R. psychosocial rehabilitation programme on the quality of care at the longer-term inpatient psychiatric departments | Inpatient Unit | Experimental not randomized ("before and after" design) | All Czech psychiatric hospitals focused on longer-term inpatients, mainly with a diagnosis of schizophrenia | 14 units for 499 patients with severe mental illness with complex needs | Other: not described |
| Yip, 2006 | China | To review and evaluate the implementation of community mental health in the People's Republic of China | CMHC | Narrative style | Inapplicable | Inapplicable | Public |
| Kleiner and Drews, 1992 | USA and Norway | To describe the experiences in the creation of innovative service delivery system which integrates psychiatric services with lay community support systems and patient social networks | CMHC | Narrative style | Psychotic patients who had more than two years of cumulative hospitalization, and who could not be placed with relatives | Inapplicable | Other: not described |
| Kraudy et al., 1987 | Nicaragua | To assess the extent to which the new proposed model had been translated into a different way of delivering psychiatric care in Nicaragua | Primary Care Centre | Descriptive Transversal | children and adult patients attending one of the surveyed services for the first time irrespective of whether or not they had a psychiatric history | 342 patients | Public |
| Lamb and Goertzel, 1977 | USA | To assess the career of psychiatrically disabled people in the community | CMHC | Descriptive Transversal | Long-term psychiatrically disable people between 18 and 64 years old who live in the community in California with a psychotic diagnoses | 99 people | Private |
| Lavoie-Tremblay et al., 2012 | Canada | To describe how families and decision-makers perceive collaboration in the context of a major transformation of mental health services and to identify the factors that facilitate and hinder family collaboration | CMHC | Thematic analysis | family members of users of mental health services and key decision makers on the mental health service | 54 family members and 22 decision-makers | Public |
| Mallik et al., 1998 | USA | To identify perceived barriers to community integration in people with psychiatric disabilities, in the areas of skills, environmental support, and community resources | Inpatient Unit | Case series | People with psychiatric disabilities in the Alliance of Psychiatric Rehabilitation Program in Baltimore County, Maryland | 42 people | Public |
| Manuel et al., 2012 | USA | To explore the experience of women with severe mental illness in transition from psychiatric hospital care to the community | Residency | Thematic analysis | women living in transitional residences on the grounds of two state-operated psychiatric hospitals in the New York City | 25 women | Public |

(*Continued*)

**Table 4.** (*Continued*)

| References | Country | Aim of study | Setting | Study design | Type of end-user or participant | Number of participants | Provider type |
|---|---|---|---|---|---|---|---|
| | | | | | metropolitan area, awaiting discharge to both supervised and independent housing in New York City | | |
| Matsea et al., 2019 | South Africa | To explore the views of different stakeholders about their roles as support systems for people with mental illness and their families in a rural setting | CMHC | Content Analysis | Stakeholders comprising traditional health practitioners (faith and traditional healer), traditional leaders, church members, home-based care team and police officers from Mashashane, a rural setting in Limpopo Province, South Africa | 41 stakeholders | Public |
| Mayston et al., 2016 | Ethiopia | To engage key stakeholders in participatory planning for a shift to mental health care integrated into primary care, and to explore their perspectives on acceptability and feasibility of the change | CMHC | Framework analysis | key stakeholders (healthcare administrators and providers, caregivers, service users and community leaders) living in Butajira town | 11 service users, 27 caregivers, 15 health extension worker and 10 health center workers | Public |
| McCubbin, 1994 | Canada | To reevaluate the recent tendency to attribute economic causes to deinstitutionalization and its subsequent "treatment in the community" mental health systems | Mixed | Narrative style | Inapplicable | Inapplicable | Mixed |
| Mechanic and Rochefort, 1990 | USA | To provide a comprehensive overview of the causes, nature, and consequences of the practice of deinstitutionalization in the United States | Mixed | Narrative style | Inapplicable | Inapplicable | Mixed |
| O'Doherty et al., 2016 | Ireland | To document the views of family members of people with an intellectual disability regarding implementation of a personalized model of social support in Ireland | CMHC | Grounded theory | parent, adult sibling or extended family member of a person receiving full-time residential supports from the agency | 40 family members | Public |
| Oshima and Kuno, 2006 | Japan | To explore how the introduction of community-based care has changed the role of psychiatric hospitals and families in caring for people with mental illness by examining the different types of living settings of clients treated for schizophrenia in Kawasaki as compared with a similar group of clients nationally | CMHC | Descriptive Transversal | adults with a diagnosis of schizophrenia living in the community and hospitalized in Kawasaki and the rest of Japan | 3.845 people living in Kawasaki and 448.000 living in Japan | Private |
| Paho, 2008 | Argentina, Brazil, Chile, Cuba, Jamaica and Mexico | To convey some of the more innovative experiences to reform mental health services implemented in Latin America and the Caribbean | Mixed | Narrative style | Inapplicable | Inapplicable | Public |

(*Continued*)

**Table 4.** (*Continued*)

| References | Country | Aim of study | Setting | Study design | Type of end-user or participant | Number of participants | Provider type |
|---|---|---|---|---|---|---|---|
| Rizzardo et al., 1986 | Italy | To analyze the impact of the reform on health care delivery by the general practitioner in an urban district in the Veneto region | Primary Care Centre | Descriptive Transversal | General practitioners working in a psychiatric service run by the University of Padua by 1983 | 24 general practitioners | Other: University facilities |
| Rose, 1979 | USA | To analyze deinstitutionalization policy on the sector of community mental health care and review its accomplishments and difficulties | Mixed | Narrative style | Inapplicable | Inapplicable | Mixed |
| Rosenheck, 2000 | USA | To review the relationship between mental health service delivery and the community in which it is embedded | CMHC | Narrative style | Inapplicable | Inapplicable | Mixed |
| Schmidt, 2000 | Canada | To examine how psychiatric rehabilitation fits within a remote First Nations community | CMHC | Thematic analysis | service providers, consumers and family members of aboriginal people with severe mental illness living in northern British Columbia | 10 stakeholders | Public |
| Semke, 1999 | USA | To explore system outcomes of interventions that were aimed at lowering high use of long-stay state hospitals | Mixed | Descriptive Transversal | adults living in the Washington state who experienced one psychiatric hospitalization of 30 days or more, or three or more psychiatric hospital admissions during a "pre-reform" period (1988) or after implementation of reform interventions (between 1991 and 1993) | 2.646.307 high utilizers of state hospitals | Public |
| Shen and Snowden, 2014 | USA | To examine whether the institutionalization of deinstitutionalization policy changed the supply of psychiatric beds in 193 countries from 2001 to 2011 | Inpatient Units | Ecological study | Mental health systems as units | 193 countries | Public |
| Stelovich, 1979 | USA | To describe factors related to deinstitutionalization leading to transfer mental health service delivery from civil mental health hospitals to prison facilities | Inpatient Unit | Narrative style | Psychiatric patients transferred to prison facilities in Massachusetts | Inapplicable | Public |
| Swidler and Tauriello, 1995 | USA | To describe the political processes leading to the Community Mental Health Reinvestment Act passage, the obstacle overcome by legislative negotiators and implementation issues | Mixed | Narrative style | Inapplicable | Inapplicable | Public |
| Sytema et al., 1996 | Italy and Netherlands | To compare the treatment of severely mentally ill patients in a community mental health service without the back-up of a mental hospital with the treatment | Mixed | Cohort study | Patient with schizophrenia that contacted a service at least once in 1988 or in 1989 in Groningen (The Netherlands) or South Verona (Italy) | 812 patients | Mixed |

(*Continued*)

**Table 4.** (*Continued*)

| References | Country | Aim of study | Setting | Study design | Type of end-user or participant | Number of participants | Provider type |
|---|---|---|---|---|---|---|---|
| | | provided in an institution-based system in which mental hospital are still predominant | | | | | |
| Wasylenki and Goering, 1995 | Canada | To describe the authors' involvement in three service delivery projects in Ontario and discuss how, by assuming multiple roles, they were able to ensure that planning and policy development were informed by current knowledge | Mixed | Narrative style | Inapplicable | Inapplicable | Public |
| Weiss, 1990 | USA | To analyze deinstitutionalization policies implemented in 1946 and 1963 in USA | Mixed | Narrative style | Inapplicable | Inapplicable | Public |
| WHO, 2014 | Malaysia, Japan, Ethiopia, Brazil, Nigeria, Uganda, UK, Iran, Italy, Portugal, Cambodia, Philippines, Spain, New Zealand, USA, Sri Lanka, Chile, India, Republic of Korea, The Netherlands, Zambia, Indonesia, Tanzania, Singapore, Lithuania, Australia, Georgia, Vietnam, South Africa, Ghana, Sweden | To capture lessons learnt from those who have been involved directly with deinstitutionalization and/or expanding community-based services and identify innovative strategies and methods associated with success of this process | Mixed | Mixed Methods | mental health experts involved directly with deinstitutionalization and/or expanding community-based services | 78 people | Public |

Abbreviations: CMHC, Community Mental Health Centre; ED, Emergency Department.

**Table 5.** Barriers to the process of psychiatric deinstitutionalization

| Category | Descriptive themes | References |
|---|---|---|
| 1. Planning, leadership, and funding | Mental health policy: Responsibility/accountability | Rose, 1979 |
| | Reform fragility: charismatic leadership | PAHO, 2008 |
| | Reform fragility: Lack of synchronization between bed reduction and development of CBMHSs | Freedman and Moran, 1984; Shen and Snowden, 2014 |
| | Reform fragility: Unaccountability of failure | Rose, 1979; Freedman and Moran, 1984; Rosenheck, 2000 |
| | Funding: Continuity of community care | Mechanic and Rochefort, 1990; McCubbin, 1994; PAHO, 2008 |
| | Funding: Hospital funds not reallocated to CMHS | Fakhoury and Priebe, 2002; PAHO, 2008 |
| 2. Knowledge/Science | Conceptual limitations and ambiguities | Bennett and Morris, 1983; Freedman and Moran, 1984; McCubbin, 1994; Mallik et al., 1998; Fakhoury and Priebe, 2002 |
| | Evidence: Lack of evidence on DI processes | Shen and Snowden, 2014 |
| | Lack of research and innovation on alternatives to institutionalization | Bennett and Morris, 1983 |
| 3. Power, interests, and influences | Irrelevance of Mental Health in the political/policy agenda | Mechanic and Rochefort, 1990; Semke, 1999; PAHO, 2008 |
| | Market factors fostering reinstitutionalization | Rose, 1979; Barton, 1983; Freedman and Moran, 1984; Dorwart et al., 1991; Fakhoury and Priebe, 2002 |
| | Uncoordinated and fragmentary advocacy actions. | Mechanic and Rochefort, 1990; McCubbin, 1994; Rosenheck, 2000 |
| | Vested interests: Pharmaceutical | McCubbin, 1994 |
| 4. Services and supports in the community | Centralized System | Kleiner and Drews, 1992 |
| | Patients: Challenging behaviors | Allen et al., 2007 |
| | Patients: Old Age | Barry et al., 2002 |
| | Services: Hospital-centric models and practices | Bennett and Morris, 1983; Kaffman et al., 1996 |
| | Early discharge | Stelovich, 1979; Kleiner and Drews, 1992 |
| | Services: Lack of services and support in the community | Weiss, 1990; McCubbin, 1994; Fakhoury and Priebe, 2002; Oshima and Kuno, 2006 |
| | Housing: Inadequate, insufficient | Mechanic and Rochefort, 1990; PAHO, 2008; Grabowski et al., 2009 |
| | Dependence on disability benefits and/or pensions | Freedman and Moran, 1984; Chopra and Herrman, 2011; Manuel et al., 2012 |
| | Patients: Disempowerment/Fatalism | Chopra and Herrman, 2011 |
| | Insufficient Public Support | Manuel et al., 2012 |
| | Patients: No money | Goering et al., 1984 |
| | Clashing views on DI within the Workforce | Kleiner and Drews, 1992; PAHO, 2008 |
| 5. Workforce | Shortages in general | Schmidt, 2000; Fakhoury and Priebe, 2002; Shen and Snowden, 2014; WHO, 2014 |
| | Shortages of specific professions | Ash et al., 2015 |
| | Inadequate training | Barton, 1983; PAHO, 2008; WHO, 2014; Mayston et al., 2016 |
| | Moral concerns and fears | Kleiner and Drews, 1992; PAHO, 2008; Ash et al., 2015 |
| | Pessimism | Cohen, 1983; Kleiner and Drews, 1992; Aggett and Goldberg, 2005 |
| | Practices of exclusion | Bryant et al., 2004; Chakraborty et al., 2011 |
| | Stigma in workforce | Barton, 1983; Semke, 1999 |
| | Vested interests: Workforce | Swidler and Tauriello, 1995; Shen and Snowden, 2014 |

*(Continued)*

**Table 5.** (*Continued*)

| Category | Descriptive themes | References |
|---|---|---|
| 6. Communities and the public | Communities are hostile toward users | Bredenberg, 1983; Fakhoury and Priebe, 2002; Aggett and Goldberg, 2005; PAHO, 2008; O'Doherty et al., 2016 |
| | Communities are ill-prepared to integrate users | Bredenberg, 1983; Fakhoury and Priebe, 2002 |
| | Public acceptance of social control | Swidler and Tauriello, 1995; Fakhoury and Priebe, 2002; Allen et al., 2007 |
| | Stigma and self-stigma | Mechanic and Rochefort, 1990; Fakhoury and Priebe, 2002; Aggett and Goldberg, 2005; ; PAHO, 2008Manuel et al., 2012; Chan and Mak, 2014; O'Doherty et al., 2016 |
| 7. Family/Carers | Broken ties between families and services | Aggett and Goldberg, 2005 |
| | Lack of support and/or unfair expectations toward families | Barton, 1983; Mechanic and Rochefort, 1990; Oshima and Kuno, 2006; ; Yip, 2006; Lavoie-Tremblay et al., 2012 |
| | Skepticism and Opposition from families | McCubbin, 1994; Oshima and Kuno, 2006 |

**Table 6.** Facilitators to the process of psychiatric deinstitutionalization

| Category | Descriptive themes | References |
|---|---|---|
| Planning, Leadership and Funding/Economic aspects | Centralized governance of the process | PAHO, 2008 |
| | Austerity and fiscal pressure | PAHO, 2008 |
| | Disability insurance | Mechanic and Rochefort, 1990 |
| | Economic incentives for DI | Mechanic and Rochefort, 1990 |
| | Fiscal strain on state mental hospital | Mechanic and Rochefort, 1990; O'Doherty et al., 2016 |
| | International policy networks and advocacy | PAHO, 2008 |
| | Intersectoral alliances and coordination | PAHO, 2008 |
| Knowledge/Science | Available evidence about alternatives | Weiss, 1990 |
| | Conceptual Clarity | Freedman and Moran, 1984; Kleiner and Drews, 1992; McCubbin, 1994 |
| | Documented Experience | Shen and Snowden, 2014 |
| | Evidence of human rights violations | PAHO, 2008 |
| | Intellectual cross-fertilization toward CBSs | Mechanic and Rochefort, 1990; PAHO, 2008 |
| | Knowledge of effects of institutions on individual patients | Bennett and Morris, 1983; Mechanic and Rochefort, 1990; Kleiner and Drews, 1992; Anderson et al., 1998 |
| | Psychopharmacological developments | Bennett and Morris, 1983; Bredenberg, 1983; Freedman and Moran, 1984; Mechanic and Rochefort, 1990; Weiss, 1990; Kleiner and Drews, 1992; Anderson et al., 1998 |
| Power, interests and influences | Human rights legislation | Anderson et al., 1998; PAHO, 2008 |
| | Influence of civil rights movements | Mechanic and Rochefort, 1990; PAHO, 2008 |
| | Legal limitations to commitment/coercion | Freedman and Moran, 1984; Mechanic and Rochefort, 1990; |
| | Legal push toward community-based treatments | Freedman and Moran, 1984 |
| | Legal standards for facility construction/operation | Anderson et al., 1998 |
| | MH Legislation | Freedman and Moran, 1984; PAHO, 2008; Shen and Snowden, 2014 |
| | Advocacy from professional organizations/groups | Weiss, 1990; WHO, 2014 |
| | International policy pressure | Shen and Snowden, 2014 |

(*Continued*)

**Table 6.** (*Continued*)

| Category | Descriptive themes | References |
|---|---|---|
| Services and supports in the community | Service-user movements and demands | Kleiner and Drews, 1992; Anderson et al., 1998 |
| | Comprehensive and structured network of CB services | Lamb and Goertzel, 1977; Cohen, 1983; Conway et al., 1994; Evans et al., 2012 |
| | Continuity of care | Sytema et al., 1996 |
| | Income for patients | Alakeson, 2010 |
| | Individualization of care in the community | Kalisova et al., 2018 |
| | Integration of mental health in PHC | Kraudy et al., 1987; PAHO, 2008; Evans et al., 2012; John et al., 2010 |
| | Limit readmission by closing beds | PAHO, 2008 |
| | Recovery-based services in a psych ICUs | Ash et al., 2015 |
| | Scale up of outpatient services | Bennett and Morris, 1983; Abas et al., 2003 |
| | Self-directed support: Autonomy in the use/selection of services | Alakeson, 2010 |
| | Shared decision-making and service user involvement | Chan and Mak, 2014 |
| | Supporting PHC expertise to raise service-user confidence | Huang et al., 2017 |
| | Social Help | Lamb and Goertzel, 1977 |
| Workforce | Anti-stigma practice | Mayston et al., 2016; Huang et al., 2017; Matsea et al., 2019 |
| | PHC training | PAHO, 2008 |
| | WF training | Weiss, 1990; Wasylenki and Goering, 1995 |
| Exogenous factors | Exogenous shocks (disasters, war) | Stelovich, 1979 |
| | Redemocratization | Rizzardo et al., 1986 |

## Workforce

Barriers related to the workforce in both institutionalized settings and community services were identified. Regarding human resources, authors mentioned staff shortages as a barrier for the transition toward community-based care (Rose, 1979; Stelovich, 1979; Fakhoury and Priebe, 2002; Shen and Snowden, 2014). Another barrier reported was the internal frictions and the existence of opposing views about care and rehabilitation (Kaffman et al., 1996; O'Doherty et al., 2016). More specifically, the psychiatric hospital workforce can delay or hinder the transformation of psychiatric institutions for fear of losing their livelihoods (Swidler and Tauriello, 1995; Shen and Snowden, 2014). Workers can express reluctance and skepticism regarding the feasibility of community living for institutionalized persons (Mayston et al., 2016; O'Doherty et al., 2016). This includes the development of unfair expectations toward family members, which alienated carers and hindered their willingness to accept responsibility (Barton, 1983).

On the other hand, service providers located in the community can be sources of stigma, expressed in the avoidance of formerly institutionalized patients (Barton, 1983), hopelessness toward treatment (Aggett and Goldberg, 2005), exclusion of users from constructing their treatment plan (Bryant et al., 2004) and fears stemming from the lack of restraining measures (Ash et al., 2015). Perceived racism at the hands of service providers can lead to mistrust in patients, causing them to either reject treatment or have poor adherence, which in turn can result in poorer outcomes, such as a longer hospital stays (Chakraborty et al., 2011).

## Communities and the public

Factors limiting social inclusion, comprising attitudes toward persons with SMI and community responses to PDI processes, were grouped under this category. Lack of preparation and stigma (Bredenberg, 1983; Mechanic and Rochefort, 1990; Fakhoury and Priebe, 2002; Aggett and Goldberg, 2005; PAHO, 2008; Manuel et al., 2012; Chan and Mak, 2014; O'Doherty et al., 2016) leads to hostile attitudes toward service-users challenging social integration (Bredenberg, 1983; Fakhoury and Priebe, 2002; Aggett and Goldberg, 2005; PAHO, 2008; O'Doherty et al., 2016). The attribution of dangerousness to individuals with SMI and the public acceptance of social control measures over recovery-oriented alternatives were also reported as barriers to PDI processes (Fakhoury and Priebe, 2002; Matsea et al., 2019).

## Family/carers

Authors highlighted the difficulties in maintaining relationships between caregivers and community services (Barton, 1983; McCubbin, 1994; Aggett and Goldberg, 2005; Yip, 2006; Lavoie-Tremblay et al., 2012; Mayston et al., 2016; O'Doherty et al., 2016). Previous experiences of failed treatments can lead to lack of cooperation and hostility toward services (Aggett and Goldberg, 2005). Professionals can be reluctant to cooperate and skeptical about the feasibility of community living (Mayston et al., 2016; O'Doherty et al., 2016). Families and caregivers may have concerns about community living and its suitability for people with high support needs (O'Doherty et al., 2016) and

concerns about receiving the burden of care, and this can alienate them and hinder their willingness to accept responsibility.

### Facilitators to the process of psychiatric deinstitutionalization

Facilitators in the process were organized under six categories summarized in Table 6 and described in detail below.

#### Planning, leadership and funding

Factors related to organizational and managerial capacities required for the transition were grouped under this category. Authors stated that the presence of a central mental health authority increased the potential to ensure effective coordination. For example, Latin America and Caribbean countries have developed mental health units within the health ministry capable of overseeing coordination (PAHO, 2008). Coordination across countries in the initial phases of reform played a crucial role, by sharing technical support and experiences of implementation (PAHO, 2008). Authors highlighted the relevance of developing intersectoral coordination, which may act as a safety net for persons with serious mental health illness reducing acute episodes (PAHO, 2008).

Studies mentioned how increases in psychiatric population and fiscal strain on state mental hospitals drove governments to develop an alternative mental health strategy (Mechanic and Rochefort, 1990; McCubbin, 1994). The pressure on fiscal resources -partly linked to economic crisis- made the costs of mental health hospitals and their inefficiency more apparent (PAHO, 2008). Also, the direct transference of funds – from reduced hospital expenditure – to community-based services was mentioned as a factor that fostered the transference of patients from state hospitals to alternative placements in the community (Mechanic and Rochefort, 1990). Finally, the growth of disability insurance was understood as a facilitator of the process of discharging service users from psychiatric hospitals by contributing to their support in the community (Mechanic and Rochefort, 1990).

#### Knowledge/science

Interdisciplinary research focusing on the legal and economic factors which influence PDI processes and practices was valued (Mechanic and Rochefort, 1990; PAHO, 2008). The elucidation of adverse effects of institutions on individual patients (Bennett and Morris, 1983; Mechanic and Rochefort, 1990; Kleiner and Drews, 1992; Anderson et al., 1998) together with the documentation of human rights violations in mental health hospitals helped in catalyzing the reform process (Bennett and Morris, 1983; PAHO, 2008). More generally, some authors stressed that conceptual clarity regarding the application of a biopsychosocial model to the mental health field (McCubbin, 1994) and the interpersonal aspect of mental health (Bennett and Morris, 1983; Kleiner and Drews, 1992) helped in the rolling up of the Deinstitutionalisation processes.

In the early stages of PDI in the USA, the allocation of research grants to state mental health hospitals developing pilot testing of outpatient treatment and rehabilitation helped in the shift of funds from mental hospitals into general hospitals (Weiss, 1990). The dissemination of early experiences of innovative policy implementation in mental health facilitated the adoption of Deinstitutionalisation practices in other regions (Shen and Snowden, 2014). Finally, the development of psychotropic medication and the reduction of psychiatric symptomatology helped to build trust in the implementation of less coercive management plans that were feasible to apply at the community level (Bennett and Morris, 1983;

Bredenberg, 1983; Freedman and Moran, 1984; Mechanic and Rochefort, 1990; Kleiner and Drews, 1992; Anderson et al., 1998).

#### Power, interests and influences

This category points to the role of social movements and organizations in influencing the development of Deinstitutionalisation processes. This includes advocacy actions and legal transformations.

Mental health professional groups and civil society organizations were seen as key agents contributing to overcome stigma and change the delivery of mental health services (Weiss, 1990). Some authors emphasized the importance of promoting the active involvement of civil society groups (Oshima and Kuno, 2006). Finally, authors highlight how the internationalization of mental health reforms puts increasing pressure on other countries to jump on the "bandwagon" to avoid appearing antiquated (Shen and Snowden, 2014).

Recognition of the rights of people with disabilities and their defense by civil rights movements fostered the development of new mental health laws promoting less restrictive therapeutic alternatives and broader transformations on mental health systems (Freedman and Moran, 1984; Mechanic and Rochefort, 1990; Anderson et al., 1998; PAHO, 2008; Shen and Snowden, 2014). These changes involved expanding the supply of options in the community (Freedman and Moran, 1984; Mechanic and Rochefort, 1990; Anderson et al., 1998; PAHO, 2008) and relocating investment from institutions to community services (Swidler and Tauriello, 1995). In some countries, an extensive and strong network of community-based organizations provided opportunities for community participation, facilitating the effective integration of patients into the community (PAHO, 2008). This was accompanied by the divulgation of reports showing mistreatment of patients in hospitals, pushing public sensitivity against asylums (Anderson et al., 1998).

#### Services and supports in the community

This category describes how the characteristics and distribution of community-based services and support for persons with SMI acted as facilitators in PDI processes.

Authors noted how policies around prevention in mental health, the integration of mental health services in primary health care centers (Kraudy et al., 1987; PAHO, 2008) and the accessibility of services (Mayston et al., 2016), together with social support such as supplementary income, can sustain community inclusion (Lamb and Goertzel, 1977), giving sustainability to Deinstitutionalisation. Adequate coordination across community-based services allowed the adequate externalization of users with complex needs (Cohen, 1983; Conway et al., 1994; Evans et al., 2012). Scaled-up outpatient facilities including local acute hospitals and intermediate facilities (Bennett and Morris, 1983; Abas et al., 2003) were key in allowing mental health systems to reduce their reliance on inpatient care and limiting beds in psychiatric settings (PAHO, 2008). Plans to end seclusion and to support mental health professionals toward a transformation in their clinical practice were identified as a facilitator to the transition (Ash et al., 2015).

Other facilitators included the continuity of care after discharge (Sytema et al., 1996) and specific actions such as: developing mobile teams and home interventions as they facilitate access to service for users who cannot physically access needed services (John et al., 2010), mitigating self-stigma dynamics by allowing an active participation of users in their treatment through shared decision-making with professional staff (Chan and Mak, 2014; Mayston et al., 2016; Matsea et al., 2019) and supporting mechanisms for

primary care workers such as a 24 h hotline for assistance when it is required (Huang et al., 2017).

In terms of training, it is argued that a reform such as PDI requires the development of an educational infrastructure including local health training networks for continuing education and training needs, and targeting providers, service-users, volunteers, family members and others (Wasylenki and Goering, 1995). The incorporation of non-specialized, community-based workers trained on mental health prevention and promotion is also highlighted (Mayston et al., 2016).

Expanding user's freedom to choose among service options was a central facilitator. This includes models of self-directed care, where users are given a budget to choose between service options (Kalisova et al., 2018). Experiences from the US, Germany and England show that patients used their budget to pay for care from their relatives, avoiding the use of institutionalized settings and preventive care options, thus shifting from crisis intervention to early interventions (Alakeson, 2010). Self-directed care improved user's autonomy and has proved to be an effective preventive intervention (Alakeson, 2010).

### Workforce

Facilitators related to community mental health services workforce were organized under this category. Strategies around training and skills include enhancing psychiatric aspects in health curriculum and provision of grants to complete training and research projects. This attracted students from other professions to the community mental health field (Weiss, 1990). Having previous experience in general medicine before training into psychiatry appeared to support a culture of community-based work and a strong collaboration with primary care teams (PAHO, 2008).

### Exogenous factors

Factors indirectly affecting the feasibility of implementing Deinstitutionalisation policies were gathered under this category. This includes the role of exogenous shocks (e.g., conflict and humanitarian disasters) (Shen and Snowden, 2014) in bringing wider public attention to patients' living conditions. A study also mentioned how the end of dictatorial regimes brought attention to human rights issues in psychiatric care, facilitating the process of Deinstitutionalisation in countries such as Argentina, Brazil and Chile (PAHO, 2008).

## Discussion

A marked decline in interest on psychiatric institutions across the global mental health literature has been noted by Cohen and Minas (2017) being absent from important prioritization exercises like the Grand Challenges in Global Mental Health (Collins et al., 2011). The authors argue that although establishing high-quality community mental health services is crucial for improving the lives of people with severe mental disorders, an exclusive focus scalability overlooks ongoing deficiencies in treatment quality and human rights protections in psychiatric institutions. Given their role in human rights abuses experienced by people with mental disorders, PDI efforts should receive more attention.

In response to this call, this article organized the available evidence around PDI, to assist in planning and conducting contextually relevant studies about and for the process. Drawing on the review, the following section introduces a set of proposals while reflecting on the limitations and problems with the available literature.

### Moving psychiatric deinstitutionalization forward

The transition from a system centered on long-term psychiatric hospital care to one centered on community-based services is complex, usually prolonged and requires adequate planning, sustained support and careful intersectoral coordination. The literature documenting and discussing PDI processes is vast, running across different time periods, regions, socio-political circumstances, and disciplines, and involving diverse models of institutional and community-based care. Based on this scoping review, we propose five key considerations for researchers and policymakers involved in PDI efforts:

1) *Needs assessment, design and scaling up.* An adequate assessment of the institutionalized population is required, to shape existing and new community-based services around their needs and preferences. A thorough analysis of the correlation of forces required to unlock institutional inertia is crucial.
2) *Financing the transition.* A comprehensive and sustainable investment is necessary, and the different aspects of the transition should be adequately costed, including new facilities, support of independent living, training, new professional roles, and the reinforcement of primary health care.
3) *Workforce development.* The workforce should be aligned with the transition from the outset. Elements such as training, incentives and guarantees of job stability are required. Curricular changes in psychiatric training, including more emphasis on community-based care and recovery-oriented practices, are necessary.
4) *PDI implementation.* The implementation process requires political resolve, careful monitoring, and an ability to respond to unexpected challenges. PDI represents a crucial learning opportunity for further scaling up.
5) *Monitoring and quality assurance.* Results of the process need to be carefully assessed against clear operational goals. The perspectives of users, caregivers, and the workforce should be incorporated into the assessments. The development of an assessment strategy detailing clear outcomes that incorporate financial and organizational dimensions is advised. Thorough documentation of PDI process, including achievements and setbacks should be done to build a reliable and diverse evidence-base for action.

A multifaceted strategy, clear and strong leadership, participation from diverse stakeholders and long-term political and financial commitment are basic elements in the planning of PDI processes. Nonetheless, implementation dynamically responds to local conditions, widely differing across countries and regions. What appears as a barrier or a facilitator can vary according to a specific context.

Although this review focuses on the barriers and facilitators for processes of PDI, we recognize that outcomes are important, and they cannot be separated from processes. Misconceptions about outcomes can hinder PDI efforts, and failed processes can lead to negative outcomes.

Two misconceptions are common. The first suggests a strong correlation between decreasing psychiatric beds and increasing homelessness or imprisonment among people with mental health problems. However, in their analysis of 23 cohort studies, Winkler et al. (2016) found that homelessness and imprisonment occurred only sporadically, and, in most studies, cases of homelessness or imprisonment were not reported.

The second misconception considers that PDI can be negative for formerly institutionalized individuals. In his review on the

impact of deinstitutionalization on discharged long-stay patients, mainly diagnosed with schizophrenia, Kunitoh (2013) found that most studies reported favorable changes in social functioning, stability and improvements in psychiatric symptoms, and positive changes in quality of life and participant attitudes toward their environment, at various time-points. Deterioration following deinstitutionalization was rare. This suggests that even long-stay patients, who commonly experience functional impairment due to schizophrenia, can achieve better functioning through deinstitutionalization.

At the same time, failure at the level of process – including planning and implementation – can lead to negative and even fatal outcomes for patients. In South Africa, from October 2015 to June 2016, a poorly executed attempt to relocate 1,711 highly dependent patients resulted in 144 deaths and 44 missing individuals (Freeman, 2018). This tragedy stemmed from ethical, political, legal, administrative, and clinical errors. Reports examining this failure offer valuable lessons for PDI efforts globally (Wessels and Naidoo, 2021).

### Limitations in the literature: Time, space, process and voice

The literature on PDI is diverse, which makes synthesis endeavors difficult. Although promoted as a global standard in psychiatric and social care, the multiplicity of contexts in which the policy has been implemented limits the possibility of finding common ground. In their systematic review of the current evidence on mental health and psychosocial outcomes for individuals residing in mental health-supported accommodation services, McPherson et al. (2018) noted how the variation in service models, the lack of definitional consistency, and poor reporting practices in the literature stymie the development of adequate synthesis.

Similarly, in a recent systematic review of psychiatric hospital reform in LMICs, Raja et al. (2021, p. 1355) expressed regret over the "dearth of research on mental hospital reform processes," indicating how poor methodological quality and the existence of variation in approach and measured outcomes challenged the extrapolation of findings on the process or outcomes of reform. Of the 12 studies they selected, 9 of them were rated as weak according to their quality assessment.

Beyond the challenges posed to synthesis efforts and through conducting this review, we identified four wider problems affecting the literature documenting PDI planning and implementation. They are related to *time*, *location*, *focus*, and *voice*.

In terms of *time*, most of the work addressing PDI was developed at the end of the 1970s through the 1980s and early 1990s. After this, there are barriers and facilitators documented which indirectly relate to the development of community-based services and their evaluation, with PDI as the "background" but not as the main object of attention. Also, the date of the search – May 2020 – could potentially exclude studies that worked with data from the pre-COVID period.

When it comes to *location*, while there is a wealth of literature on the topic, it is important to note that much of it is based on the experiences of the USA and Western Europe. The documentation of PDI in regions outside of the "global north" is typically limited to personal testimonies from process leaders, which may lack systematicity and are usually published in languages other than English. This can restrict their accessibility and dissemination.

In terms of *focus*, most studies have a clinical orientation, evaluating various outcomes that are directly or indirectly related

to PDI. However, the process itself, has received little attention. An exclusive emphasis on outcomes can obscure the administrative, legal, and political complexities of carrying out a psychiatric reform, this, hinder the dissemination of important lessons.

Finally, it is worth noting that important *voices* are often missing from available studies and reports on PDI processes. While some studies do consider the experiences and engagement of caregivers, healthcare workers, and patients, they are still in the minority. This can create a skewed understanding of the impact of PDI, as these individuals play crucial roles in shaping the process and its outcomes. The same goes for the different communities where patients have developed their lives after PDI.

These limitations have significant consequences. It is unclear whether the evidence extracted from experiences in high-income countries in North America and Europe can directly inform processes in other regions, including low- and middle-income countries (LMICs). While it is possible to identify common pitfalls, barriers, and needs, this identification must be accompanied by up-to-date local research to ensure that the evidence is relevant and applicable to specific contexts.

The involvement of patients and communities affected by institutionalization in the design and implementation of research and policy should be central in a renewed PDI agenda. The recently launched Guidelines on deinstitutionalization, including in emergencies, by the United Nations Committee on the Rights of Persons with Disabilities represent a pioneering effort in this direction (OHCHR, 2022).

At the same time, qualitative and ethnographically oriented case studies are required to closely examine PDI efforts while remaining attentive to diversity and local creativity beyond global normative parameters of success and failure. Furthermore, reflexive, and flexible approaches to research synthesis are necessary to capture and assess the wealth of lessons learned from diverse engagements with deinstitutionalization across the globe.

This article offers a preliminary and general classification of barriers and facilitators that can inform the development of relevant research through various methodologies and other literature. The categories can be modified and customized based on the evidence from various settings. As far as we know, this classification is not yet present in the existing literature.

### Conclusion

Institutional models of care continue to dominate mental health service provision and financing in many countries, leading to a continued denial of the right to freedom and a life in the community for individuals with mental health conditions and associated disabilities. The successful implementation of PDI requires detailed planning, sustained support and coordinated action across different sectors.

This review identifies the factors impacting PDI processes, according to the available literature. Barriers and facilitators are organized in 15 thematic groups. The results reveal that PDI processes are complex and multifaceted, requiring detailed planning and commensurate financial and political support. We have offered five considerations for policymakers and researchers interested and/or involved in PDI efforts.

There are many lessons to be learned from the processes described in the literature, and many areas where research has been insufficient. Barriers and facilitators will differ in response to the legal, institutional, and political characteristics of each region and

country. This categorization can be adapted to national realities and different levels of policy progress in PDI, to guide research and policy efforts. We call for methodological innovation and the involvement of affected communities as key elements of a renewed research agenda around this neglected aspect of mental health reform worldwide.

**Open peer review.** To view the open peer review materials for this article, please visit http://doi.org/10.1017/gmh.2023.18.

**Supplementary material.** The supplementary material for this article can be found at https://doi.org/10.1017/gmh.2023.18.

**Data availability statement.** The authors confirm that the data supporting the findings of this study are available within the article (and/or its Supplementary Materials).

**Author contribution.** M.I.D. and C.M.C. conceived the idea for the project. J.U.O. and C.M.C. developed the framework to conduct the systematic search, which J.G.M. performed. J.U.O. and J.G.M. established the eligibility of articles under the supervision and with the contribution of C.M.C. J.G.M. and J.U.O. extracted the data of the selected articles. J.G.M., J.U.O. and C.M.C. coded the article contents and created the categories iteratively through rounds of revision and adjustment. J.U.O. and C.M.C. produced an early draft of the manuscript. F.T. reviewed several versions of the manuscript. The final manuscript was discussed and improved by all the authors. C.M.C. and J.U.O. coordinated the development of the manuscript.

**Competing interest.** The authors declare none.

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
