## [Reviewer Report · Review: Moving psychiatric deinstitutionalisation forward: A scoping review
of barriers and facilitators. — R0/PR2]

This review aimed to assess barriers and facilitators related to psychiatric
deinstitutionalization, i.e. shifting the locus of mental health care from large
psychiatric hospitals to the community. The paper is topical, since deinstitutionalization
is a conditio sine qua non of adherence to human rights as defined by CRPD; it makes sense
also clinically and economically; and it has not been fully or even partially achieved in
many countries across the globe, which is why it remains as one of the main aims of WHO
policy, especially in the region of central and eastern Europe. The review seems to be
well conducted and clearly reported; however, there are some points that I would like to
see addressed:

Major issues:

-1) I am afraid that quite a few important references could have been missed because:

o Only three electronic databases were searched and some important ones, such as PsycINFO
or Web of Science were omitted.

o The original search was done in May 2020, which is more than 2.5 years back (covid-19
justification does not hold, since many studies published after May 2020 work with data
from the pre-covid period).

o Reviews selected to inform the search strategy do not include the major relevant ones
for psychiatric deinstitutionalization:

- Kunitoh, N., 2013. From hospital to the community: The influence of
deinstitutionalization on discharged long‐stay psychiatric patients. Psychiatry and
clinical neurosciences, 67(6), pp.384-396.

- Winkler, P., Barrett, B., McCrone, P., Csémy, L., Janous̆ková, M. and Höschl, C., 2016.
Deinstitutionalised patients, homelessness and imprisonment: systematic review. The
British Journal of Psychiatry, 208(5), pp.421-428.

- Both of the abovementioned reviews included original studies on discharged psychiatric
patients. These studies might contain many important lessons (or barriers and
facilitators) that might be important for the current paper.

- I do not want authors to re-run their search and screening, but I believe they should
make an effort to reduce possible bias stemming from the above-mentioned points.

- 2) It is fine not to assess the methodological quality of studies included in the final
analysis of a scoping review. However, authors should be critical of the evidence provided
and not to present claims as facts, unless there is a decent amount of evidence to support
them. For instance:

o Obviously, the pharmaceutical industry supports a pharmacological research that could
be potentially of an economic benefit to pharmaceutical companies, but that does not mean
that it is intentionally „shifting the focus of research away from community-based
therapeutic and rehabilitative approaches“, does it? Is there any empirical evidence for
that?

o Similarly, what is the empirical evidence for the claim that a revolving door
phenomenon is perpetuating stigma and discrimination against mental health users?

o Or the following: „In places like the USA, at the early stages of PDI, many patients
experienced further crises’ driving them to a new psychiatric hospitalization or directly
into prison (Shen & Snowden, 2014). This was related to inadequate training of
community-based workers, discharge without community support (Shen & Snowden, 2014)
and early release promoted by legislatively mandated Deinstitutionalisation policies
(Kleiner & Drews, 1992).“ Here, the referred study is based on correlations of
cross-sectional data which is prone to ecological fallacy, and the conclusions might
simply not be true, please read the above-mentioned review on deinstitutionalised
patients, homelessness and imprisonment.

- please make it sure that unsupported claims made elsewhere are not blindly repeated in
this paper

- 3) Discussion needs to contain a section, where findings or thoughts published in other
relevant studies are discussed.

Minor issues:

- Unstructured abstract should be provided

- „Progress towards PDI varies greatly across and within countries (Goldman et al., 1982;
Hudson, 2019).“ – The paper from 1982 is now outdated; there are much newer papers citable
in this regard, look them up and update citations here.

- Doublecheck the references throughout the paper and correct where appropriate (e.g.
replace „Taylor Salisbury et al., 2016“ with „Salisbury et al., 2016“)

- I believe that direct citations require the page number to be included in the reference
(i.e. Mays et al., 2001 is not enough).

- „Institutional models of care continue to dominate mental health service provision and
financing in many countries, leading to a continued denial of the right to freedom and a
life in the community for millions of individuals with mental health conditions and
associated disabilities.“ Do you want to say here that millions of psychiatric inpatients
are unnecessarily institutionalized? Can I see some evidence to justify the use of
„millions“ here?

---

## [Reviewer Report · Review: Moving psychiatric deinstitutionalisation forward: A scoping review
of barriers and facilitators. — R0/PR3]

The article presents a comprehensive literature review and looks for factors that
facilitate and/or hinder the deinstitutionalization of psychiatric care. The authors
review English and Spanish language essays related to the deinstitutionalization process
around the world. The topic is beneficial because many countries still need to complete
deinstitutionalization, and many are still waiting for this step. A limitation but on the
other side strength of the review is the more than 40-year evaluation period (articles
from 1977-2019) and the vastness and diversity of the monitored area (the whole world).
Nevertheless, the results of the presented study can be considered summarizing and
essential. Therefore, I recommend adding a paragraph in the introduction about the
countries in which deinstitutionalization took place successfully.

---

## [Reviewer Report · Review: Moving psychiatric deinstitutionalisation forward: A scoping review
of barriers and facilitators. — R1/PR7]

Thank you for improving your paper by responding to the points being raised. However, I
would like to see more discussion related to the outcomes of PDI processes. I understand
that your study is focused on processes rather than on outcomes, but these two are not
completely separable, and the article would greatly benefit if you could discuss both, a)
generally positive outcomes of PDI processes as reported in relevant systematic reviews*,
and b) cases of PDI processes that turned into unfavorable outcomes for patients, most
notably the recent case of South Africa**, and to do this in light of your findings. After
all, this is being already touched upon in the paragraph starting with “These findings are
consistent with other reviews. In their review on deinstitutionalization and the ”home
turn“ from the 1990s...”

*

Winkler, P., Barrett, B., McCrone, P., Csémy, L., Janous̆ková, M. and Höschl, C., 2016.
Deinstitutionalised patients, homelessness and imprisonment: systematic review. The
British Journal of Psychiatry, 208(5), pp.421-428.

Kunitoh, N., 2013. From hospital to the community: The influence of
deinstitutionalization on discharged long‐stay psychiatric patients. Psychiatry and
clinical neurosciences, 67(6), pp.384-396.

**

Freeman, M.C., 2018. Global lessons for deinstitutionalisation from the ill-fated
transfer of mental health-care users in Gauteng, South Africa. The Lancet Psychiatry,
5(9), pp.765-768.